# Interlinkages of Water-Related SDG Indicators Globally and in Low-Income Countries

**Andi Besse Rimba *** and **Yukiko Hirabayashi**

Department of Civil Engineering, Shibaura Institute of Technology, 3-7-5 Toyosu, Tokyo 135-8548, Japan
* Correspondence: arimba@shibaura-it.ac.jp

**Abstract:** The international community has committed to protecting the Earth and its ecosystems, thus ensuring wellbeing, economic growth, and a sustainable environment, by applying 17 sustainable development goals (SDGs), including many related to water. These goals and their indicators can have synergistic, trade-off, or neutral interlinkages. This study measured the interlinkages between 31 SDG indicators directly or indirectly related to water belonging to seven categories: extreme water events, water availability, water quality and waterborne diseases, energy-related water, industry and technology-related water, water governance and management, and ecosystem-related water. All the indicators were paired, resulting in 450 pairs. The interlinkage between water-related indicators globally and in low-income countries (LIC) were determined by Spearman's rank correlation ($\rho$), and standardized multilinear regression was applied to identify the dominant drivers of synergistic and trade-off interactions. The finding shows that water quality, waterborne disease, and energy-related water are the most feasible to achieve in SDGs in global and LIC. The local government may take advantage from this study. Moreover, the government should pay attention in developing and providing alternative energy especially in LIC due to some trade-offs appeared with health and social conflict may arise. The interaction between indicators become the main driver of synergy/trade-off over population and GDP in interlinkage water related SDGs.

**Keywords:** water; health; energy; trade-off; synergy; SDGs

## 1. Introduction

Water is inextricably associated with the development of all nations. However, improper water management and unsustainable targets of national development are causing pressure on water resources, especially for low-income countries (LIC). In 2017, approximately 2.2 billion people suffered from unsafely managed water services (i.e., a 30 min walk to access clean water, contaminated wells and springs, and polluted surface water) [1]. Poor water quality and sanitation are connected with the transmission of water-related diseases, e.g., cholera, typhoid, malaria, dysentery, hepatitis A, polio, and neglected tropical diseases (NTDs) [2]. In 2017, mortality due to unsafe water accounted for 2.2% of global deaths, and 6% of deaths in low-income countries [3].

In September 2015, the leaders of one hundred ninety-three countries agreed to adopt the United Nations sustainable development goals (SDGs), which aim to reduce poverty to ensure better quality of life and a sustainable environment by 2030. There are interdependencies and interactions between the SDGs; however, the outputs of the SDGs may be parallel or opposite [4]. The interlinkages between SDGs are considered synergistic (where the progress of one goal contributes to the success of other goals), trade-offs (where the progress of one goal impedes the progress of another), or neutral (where two indicators have limited associations). For all countries to achieve all SDG goals within funding and resource constraints, it is necessary to prioritize the SDGs target that is possible to achieve especially in Low Income Countries (LIC). Hence, interlinkage (synergies and trade-offs) between SDG indicators should be clearly identified. Hence, the local government in global

region and in LIC may take advantages from this study to priorities their goal in achieving SDGs especially all goals related to water.

Some researchers have investigated the interlinkage of SDG indicators globally and at the country level by empirical analysis [5–8] and the literature investigation [9], but quantitative and regionally detailed synergies and tradeoffs are needed. The previous study had investigated that positive relationships between indicator pairs were identified to outweigh the negative association in most countries [5]. Moreover, synergy always outweighs trade-off in SDGs interaction considering population, location, income, and regional group, especially in SDG 1, 5, and 6 [10]. However, the dominant drivers of increases and decreases in SDG indicators is not well discussed.

Water is associated with many sectors and affects the increases and decreases in production of sectors [11]; water stands out as the most critical sustainable development challenge since it deals with the most valuable and limited resource on earth. If water becomes scarce or threatened, it causes risks to economic, social, and environmental sustainability [12]. Moreover, water also influences the increases and decreases in SDG indicators, as demonstrated in the literature on SDG interlinkages [8]. The limitation and availability of water or water-related disasters is an important key to proceed SDGs, as water relates to several many SDG targets in many contexts. Furthermore, the latest sixth Assessment Report on Intergovernmental Panel on Climate Change reported that climate change has negatively impacted many water use sectors [13]. Hence, the government needs a practical guideline to achieve the SDGs with clear information on the interlinkage of water-related SDG targets. Additionally, such information could counter the impact of water-related to societies, and help the government issue regulations, Especially in LIC face more obstacles in achieving SDGs due to limitation of funding, technology, and capability [8].

This study aims to address the interlinkages of SDGs directly and indirectly related to water resources, and determine the extent to which the interlinkages can be measured in global and LIC. Thus, this study measured the achievement of SDGs globally and in LIC. Moreover, the role of water in achieving SDGs and its relation to other goals were examined. The empirical research on SDG interlinkages may contribute to developing water-related policies globally and in LIC.

## 2. Materials and Methods

### 2.1. Study Area

SDG indicators in global (for all continents) and 26 low-income countries (LIC) were analyzed as the World Bank classification [14]. Low-income countries have a gross national income (GNI) per capita of $1025 or less according to the World Bank classification (Figure 1).

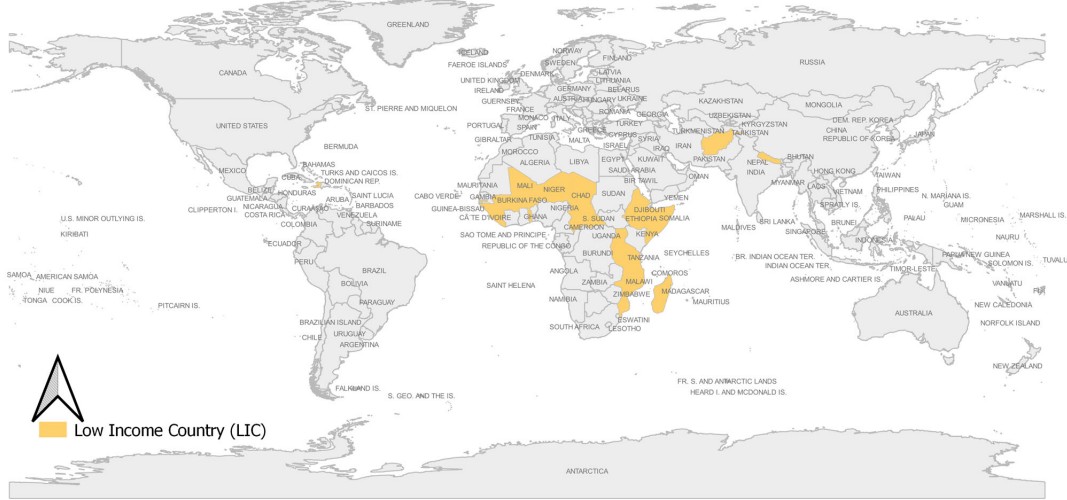

**Figure 1.** Study area. Asia: Afghanistan (AFG), Nepal (NPL). Africa: Central African Republic (CAF), Republic of Chad (TCD), Republic of the Congo (ZAR), Republic of Burundi (BDI), Ethiopia (ETH),

State of Eritrea (ERI), Republic of Guinea (GIN), Republic of Liberia (LBR), Republic of Madagascar (MDG), Republic of Malawi (MWI), Republic of Mali (MLI), Republic of Mozambique (MOZ), Niger (NER), Rwanda (RWA), Republic of Seychelles (SLE), Somali Republic (SOM), South Sudan (SSD), Togolese Republic (TGO), Republic of Uganda (UGA), United Republic of Tanzania (TZA), Burkina Faso (BFA), Republic of the Gambia (GMB), Republic of Guinea-Bissau (GNB). South and Central America: Republic of Haiti (HTI).

*2.2. Data*

Data were retrieved from international institutions that provide global data. The United Nations Statistics Division provides data from 1983 to 2018 for 122 SDG indicators (https://unstats.un.org/sdgs/indicators/database, accessed on 3 September 2021). The World Bank provided GDP per capita from 1960 to 2020 (https://databank.worldbank.org/source/world-development-indicators, accessed on 3 September 2021). Global population data were downloaded from the United Nations Population Division (https://population.un.org/wpp/Download/Standard/Population/, accessed on 3 September 2021).

The primary water-related SDG is Goal 6 (Clean Water and Sanitation). However, Goal 1 (No Poverty) [15], Goal 2 (Zero Hunger) [16], Goal 3 (Good Health and Well-being) [17], Goal 4 (Quality Education) [18], Goal 7 (Affordable and Clean Energy) [19], Goal 9 (Industry, Innovation and Infrastructure) [20], Goal 11 (Sustainable Cities and Communities) [21], Goal 12 (Responsible Consumption and Production) [22], Goal 13 (Climate Actions) [23], Goal 14 (Life Below Water) [24], and Goal 15 (Life on Land) utilize, or are directly or indirectly influenced by water management, quantity, and quality. From 122 SDG indicators in the UN SDG database, 33 direct or indirect indicators was selected (Table 1) that covered seven water-related categories: A. extreme water events, B. water availability, C. water quality and waterborne disease, D. water related to energy, E. water related to industry and technology, F. water governance and management, and G. water related to ecosystems.

**Table 1.** List of the selected water-related SDGs.

| No. | Indicator | Description | Category | References | UN Data Trend (P = Positive/N = Negative) |
|---|---|---|---|---|---|
| 1 | 1.5.1 | Number of deaths and missing persons | A | [25,26] | N |
| 2 | 2.3.1 | Production per labor unit by classes of farming/pastoral/forestry sectors farming/pastoral/forestry enterprise | B | [27] | P |
| 3 | 2.3.2 | Average income of small-scale food producers | B | [28] | N |
| 4 | 2.4.1 | Agricultural area under productive and sustainable agriculture | B | [29] | No data |
| 5 | 3.3.3 | Malaria | C | [30,31] | N |
| 6 | 3.3.5 | Number of people requiring interventions against neglected tropical diseases | C | [32] | N |
| 7 | 3.9.2 | Mortality rate attributed to unsafe water, unsafe sanitation and lack of hygiene | C | [33] | N |
| 8 | 4.a.1a | Schools with drinking water | B, C | [2] | P |
| 9 | 4.a.1b | Schools with basic handwashing | B, C | [34] | P |
| 10 | 6.1.1 | Population with safe drinking water | C | [35,36] | P |

**Table 1.** *Cont.*

| No. | Indicator | Description | Category | References | UN Data Trend (P = Positive/N = Negative) |
|---|---|---|---|---|---|
| 11 | 6.3.1 | Proportion of wastewater safely treated | C, G | [37] | P |
| 12 | 6.3.2 | Proportion of water bodies with good ambient water quality | C | [38] | P |
| 13 | 6.4.1 | Change in water-use efficiency over time | B, F | [36,39] | P |
| 14 | 6.4.2 | Water stress level: proportion of water withdrawal and freshwater | B, F | [40] | P |
| 15 | 6.6.1a | Nationally derived extent of open water bodies | B, F | [41,42] | N |
| 16 | 6.6.1b | Water body extent | B, F | [41] | N |
| 17 | 6.a.1 | Amount of water- and sanitation-related official development assistance | C, F | [43,44] | P |
| 18 | 6.b.1 | Communities with water and sanitation management supported by government | F | [45] | P |
| 19 | 7.1.1 | Proportion of population with access to electricity | D | [46,47] | P |
| 20 | 7.1.2 | Proportion of population with primary reliance on clean fuels and technology | D | [48] | P |
| 21 | 7.2.1 | Renewable energy share in the total final energy consumption | D | [49] | N |
| 22 | 9.5.1 | Research and development | E | [20] | P |
| 23 | 11.3.2 | Cities with a direct participation structure of civil society in urban planning | F | [50] | No data |
| 24 | 11.5.1 | Number of deaths and missing persons | A | [25,26] | N |
| 25 | 12.2.2 | Domestic material consumption | B, E | [11] | P |
| 26 | 12.4.2a | Hazardous waste treated or disposed (%) | C, G | [51] | P |
| 27 | 12.4.2b | Electronic waste recycling (%) | C, G | [52] | P |
| 28 | 13.1.2 | Number of deaths and missing persons | A | [25,26] | N |
| 29 | 14.3.1 | Average marine acidity (pH) (agreed suite of representative sampling stations) | G | [53] | P |
| 30 | 15.1.1 | Forest area as a proportion of total land area | G | [54] | N |
| 31 | 15.1.2 | Terrestrial and freshwater biodiversity in protected areas | G | [55] | P |
| 32 | 15.3.1 | Degraded land over total land area | G | [56,57] | P |
| 33 | 15.5.1 | Red List index | G | [58] | N |

Notes: Category: A. extreme water events, B. water availability, C. water quality and waterborne disease, D. water related to energy, E. water related to industry and technology, F. water governance and management, and G. water related to ecosystems. P is the trend data of observing year tend to increase, while N shows vice versa of P.

Table 1 shows the selected indicators, grouped into seven categories, and associated references and historical trends. A positive (P) historical trend indicates that the actual value of the indicator increased during the observation period; for example, indicator 6.1.1 (Supplementary Figure S1a) shows that the population with access to safe water has increased. A negative (N) historical trend indicates that the actual indicator value tended to fall (Supplementary Figure S1b). The positive and negative historical trends in Table 1 are not representative of SDG milestones, they represent observed values, which

may decrease or increase depending on other factors. Details of past change indicators for all income levels, i.e., high-income countries (HIC), upper–middle-income countries, and lower-middle-income Countries (MIC), and low-income countries (LIC) are given in (Supplementary Figure S2), and a scatter plot of sample data is shown in (Supplementary Figure S3). Figures S2 and S3, the increasing/decreasing of graph may influenced by GDP or population for each year. Hence, we need to remove the effect of GDP and population by averaging the data.

Of the 33 water related SDG indicators, 2 lacked data and were removed from the calculations (indicators 2.3.1 and 14.3.1). Hence, a total of 31 indicators and their combinations were analyzed. To prevent double counting in the statistical calculations (spearman's rank), each pair was analyzed only once, with the assumption that the combination of indicator A and indicator B is the same as the combination of indicator B and indicator A, and we excluded combination of indicator A and indicator A. Thus, a total of 450 pairs of indicators were included in the analysis. Improvements in past indicators are often highly correlated with past economic development; for example, in Africa, malaria decreased by 10% as the GDP increased by 0.3% [59]. As shown in Supplementary Figure S4, the increase of 3.3.5 and 7.2.1 has a high correlation with the increasing number of populations with $R^2$ 0.869 and 0.994, respectively; moreover, the increase in indicator 9.5.1 shows a high correlation ($R^2 = 0.800$) with increasing GDP. Therefore, we first averaged each country's indicators over the study period and then analyzed the relationships between pairs of indicators. We analyzed only those data that covered a minimum of 20 countries for global data and 10 countries for low-income countries to reduce the number of statistical calculations and obtain meaningful results.

### 2.3. Method

2.3.1. Extracting Synergies and Trade-Offs through Statistical Logic: Spearman's Rank Correlation ($\rho$)

Spearman's rank calculation was utilized to find the class, i.e., synergisms, trade-offs, or neutral associations, of indicator pairs [5]. Spearman's rank is always between −1 and +1; pairs with $\rho$ values > 0.5 were categorized as synergisms (i.e., positive associations), those with $\rho$ values < −0.5 were categorized as trade-offs (negative associations), and those with $\rho$ values from −0.5 to 0.5 were not classified (to avoid over-interpretation) [10]. However, an analogous rationale would apply for pairs classified as synergisms and trade-offs [5]. The Spearman's rank correlation was calculated with Equation (1):

$$\rho = 1 - \frac{6 \sum d_i^2}{n(n^2 - 1)} \tag{1}$$

where:

$\rho$ = spearman's rank correlation coefficient;
$d_i$ = the difference between the ranks of two SDG indicators;
$n$ = the number of time series pairs in one pair of indicators.

Multiple linear regression (MLR), known as multiple regression, is the extension of ordinary least-squares (OLS) regression. MLR is a statistical technique that uses several explanatory variables to find the linear relation between the explanatory (independent) variable and response (dependent) variable.

$$y_i = a_0 + a_1 x_{i1} + a_2 x_{i2} + a_3 x_{i3} + \varepsilon \tag{2}$$

$y_i$: the 1st SDG indicator of the pair for country $i$ (dependent variable);
$a_0$: y intercept (constant term);
$a_1$: slope coefficient for the 2nd SDG indicator of the pair for country $i$;
$x_{i1}$: the 2nd SDG indicator of the pair for country $i$ (independent variable);
$a_2$: slope coefficient for the GDP of the country $i$;
$x_{i2}$: GDP of country $i$ (independent variable);

$a_3$: slope coefficient for the population of country $i$;

$x_{i3}$: population of country $i$ (independent variable);

$\varepsilon$: the model's error term (residual).

The coefficient of determination (R squared) is a statistical metric utilized to measure how much of the variation in the dependent variable can be explained by the variation in the independent variables. The value of $R^2$ is between 0 and 1, where 0 means no correlation, and one indicates that the prediction generated by the independent variables has no error.

2.3.2. Identifying Influential Variables: Magnitude of the Standardized Coefficient (Beta) in the Multiple Linear Regression

It can be difficult to compare the regression coefficients of variables with different units; a small coefficient may be more important than a larger one. Standardizing regression coefficients eliminates this problem by expressing the coefficients in a single, standard set of statistically reasonable units so that comparison may at least be attempted. The regression coefficient $b_i$ indicates the effect of a change in $X_i$ on $Y$, maintaining all of the other $X$ variables unchanged. The measurement units of the regression coefficient $b_i$ are the units of $Y$ per unit of $X_i$.

The standardized regression coefficient, found by multiplying the regression coefficient bi by $S_{x_i}$ and dividing the result by $S_Y$, represents the expected change in $Y$ (in standardized units of $S_Y$, where each "unit" is a statistical unit equal to one standard deviation) due to an increase in $X_i$ of one of its standardized units (i.e., $S_{x_i}$), maintaining all other $X$ variables unchanged. The absolute values of the standardized regression coefficients may be compared, giving a rough indication of the relative importance of each variable. Each standardized regression coefficient is expressed in units of standard deviations of $Y$ per standard deviation of $X_i$ [60]. Each regression coefficient is adjusted according to a ratio of ordinary sample standard deviations. The absolute values give a rough indication of the relative importance of the $X$ variables, as shown in Equation (3).

$$b_i \frac{S_{X_i}}{S_Y} \tag{3}$$

A standardized coefficient identifies the dominant variable that influences the indicators. Standardization must be performed since the units of all the variables are different. The regression was calculated twice to determine the consistency of the magnitude of the results. The dependent variables were indicators 1 or 2, and the independent variables were indicator 1, indicator 2, GDP, and population, as shown in Equation (2). Most pairs depended on the correlation between indicator one and indicator two. The standardized coefficients can be seen in Supplementary Table S1.

## 3. Results

### 3.1. Interlinkages between Water-Related SDG Indicators in Global

The total number of interlinkages (i.e., synergisms, trade-offs, and neutral associations), and the total number of indicator pairs for each goal are shown in Figure 2. These results show more neutral interlinkages than synergistic or trade-off interlinkages; Goals 3 and 6 showed the highest synergies, while goal 7 showed the highest trade-offs compared to the other goals. Goal 6 has a more significant number of pairs. Interlinkage in global analysis show more neutral as an impact of a cross-section analysis [10].

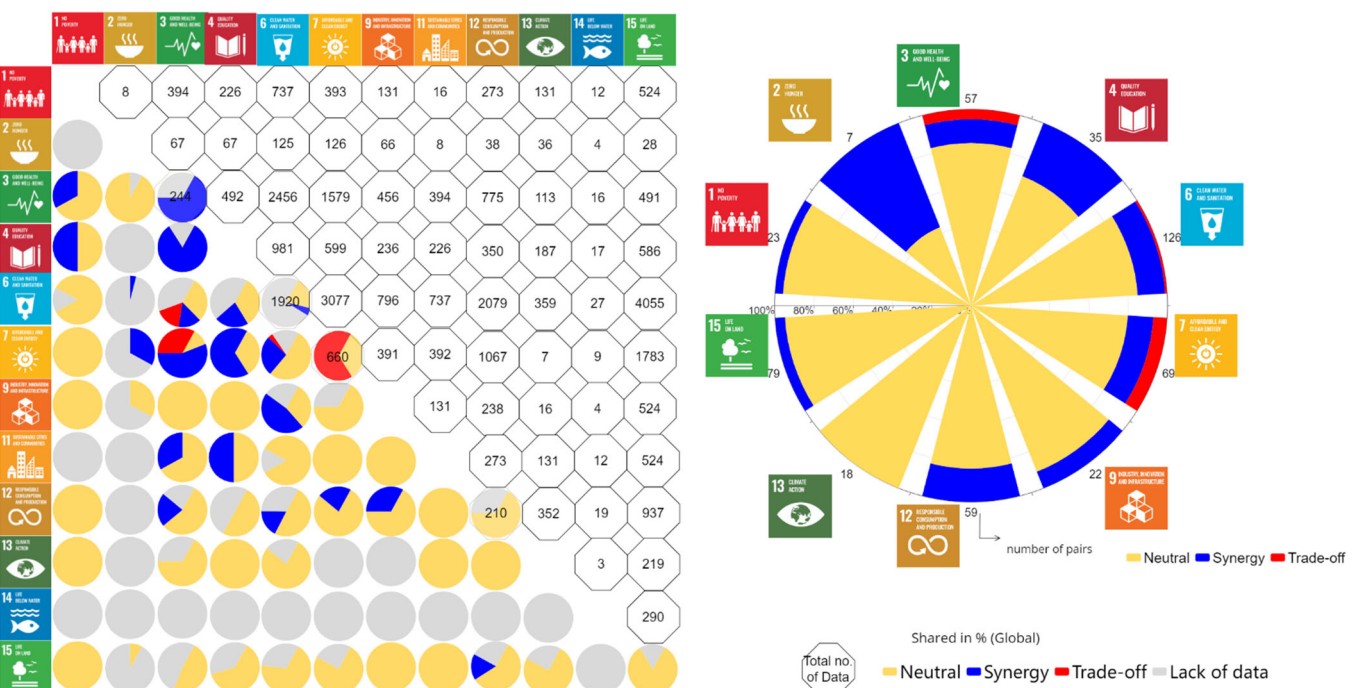

**Figure 2.** Matrix of interlinkage percentage with number of samples (**left**) and shares in percentage of global interlinkage with total number pairs (**right**).

### 3.1.1. Water Facilities in Schools

Water quality and access to clean water at home and school (4.a.1) is an indicator of goal 4 and rank first in terms of shared in approximately 25% of synergy in goal 4 (Figure 2). Supplementary Table S2 shows that schools with drinking water (4.a.1a) and school with basic handwashing (4.a.1b) synergize with changes in water-use efficiency over time (6.4.1) with $\rho = 0.60$ and $\rho = 0.55$, respectively; moreover, indicator 4.a.1b synergies with a population with safe drinking water (6.1.1) with $\rho = 0.79$. Billions of people globally have gained access to basic services, water, and sanitation since 2000, and the number of schools with drinking and sanitation facilities has increased [1]. However, the growth percentage of basic services in homes and schools does not mean that communities worldwide have adequate basic water services. Many schools in the U.S. deal with aging infrastructure and consequent water safety problems, and rely on bottled water delivery systems [61]. Moreover, according to a report from the WHO and UNICEF, there are significant gaps in data on the effectiveness of monitoring inequalities in Adequate water, sanitation, and hygiene (WaSH), and only 35%, 48%, and 30% of the global population reported data on the safe management of drinking water, sanitation, and handwashing, respectively [1].

The synergy between water and education was influenced by GDP and the correlation between indicators. GDP influenced the synergy between 4.a.1a and 6.4.1 (water-use efficiency), with a standardized coefficient of 0.49 (in Supplementary Table S1). The increase in GDP due to technology, such as automatic faucets, increases water-use efficiency. On the other hand, the increase in the proportion of the population with drinking water is greater than the percentage of the population with water facilities in schools, as shown in Supplementary Table S1. The population with drinking water (6.1.1) has a more significant influence on handwashing facilities at school (4.a.1b) than GDP (standardized coefficient 0.01), and population (standardized coefficient 0.07), with a standardized coefficient of 0.81 (in Supplementary Table S1).

### 3.1.2. Health and Water

As shown in Figure 2, more than 40% of goal 3 shows synergy and trade-off with 57 total number of pairs. Reducing the number of deaths due to malaria (3.3.3), NTDs

(e.g., ascariasis, Buruli ulcer, Chagas disease, dracunculiasis, hookworm infection, human African trypanosomiasis, leishmaniasis, leprosy, lymphatic filariasis, onchocerciasis, schistosomiasis, trachoma, and trichuriasis) (3.3.5), and unsafe water and sanitation (3.9.2) are the target of goal 3. Indicator 3.3.5 showed a strong synergistic interlinkage with water-use efficiency (6.4.1) ($\rho = 0.58$), as shown in Supplementary Table S2. A well-managed water system increases the efficiency of water use, especially in drought conditions (i.e., periods with low water availability and high-water demands), and in rainy seasons (i.e., periods with high flood potential).

Water- and health-related indicators show more synergies with other sectors, and energy-related indicators show trade-offs with some sectors. Increasing piped water access and improved sanitation coverage in poor communities, for example in Indonesia, reduced the prevalence of diarrhea (Figure 3). However, renewable energy-related indicators show greater trade-offs than indicators of other sectors since renewable energy requires deeper research and increased investment. Nevertheless, some countries target reducing fossil fuel and increasing renewable energy shares in their total energy consumption (7.2.1); for instance, Indonesia plans to achieve 23% and 31% renewal energy by 2025 and 2050, respectively, to meet the goals of the Paris climate agreement and contribute to greenhouse gas reduction. The Indonesian government educates and encourages communities, especially those in rural areas, to transform animal waste to biogas energy for cooking; however, odors from biogas plants can affect the local environment if the gases are not treated properly. In addition, the high capital cost for installing biogas plants and misconception regarding the reliability of biogas plants present obstacles.

Furthermore, droughts reduce water quality and create health risks due to the potential for increased absorption of groundwater contaminants [62]. When people consume polluted groundwater, their health may be affected. In addition, during the rainy season, flooding may catalyze an increase in diarrheal and water-related diseases. Moreover, in this study, we found that mortality due to unsafe water and sanitation (3.9.2) was synergetic with the percentage of populations with drinking water (6.1.1) ($\rho = 0.79$), as shown in Supplementary Table S2. Increasing the availability of drinking water in a community reduces mortality due to unsafe water and sanitation. This finding aligns with WHO and UNICEF data showing that 2 billion and 1.8 billion people globally, respectively, have been given increased access to improved drinking water sources and improved hygiene. Moreover, this finding shows that increasing water access reduces the spread of NTDs. A concrete example can be seen in Figure 3. The percentage of drinking water coverage increased 6.71% from 2007 to 2014, and the number of deaths due to unsafe water decreased by up to 2448 people in Indonesia.

This study found (in Supplementary Table S2) that there were trade-offs between health indicators (3.3.5 and 3.9.2) and renewable energy indicators (7.2.1). Resource reservoirs may cause health problems; for example, dams and lakes provide breeding sites for Anopheles mosquitoes, which transmit *Plasmodium falciparum*, leading to cases of malaria. Moreover, water management facilities, such as dams, ponds, irrigation, and public toilets, lead to stagnant water, creating areas that support the growth of bacteria and vectors of dengue. Furthermore, water sources may contain inorganic chemicals from a variety of geological structures [63] and human activities [24]. In tropical regions, 80% of diseases are transmitted by germs in the water [64]. Atangana 2021 found that groundwater in South Africa contains heavy metals [65]. Van Abel 2018 stated that in South Africa, the frequency of virus exposure (with a range of 10−4–1011 viruses/L) from water, especially drinking, ground, irrigation, surface, and waste waters, is extremely high [66]. Adverse health risks, including those from NTDs, are associated with these water risks. In sub-Saharan Africa, humans derive their livelihoods from wetland, which are often over-utilized, leading to human exposure to disease-causing infectious agents [17].

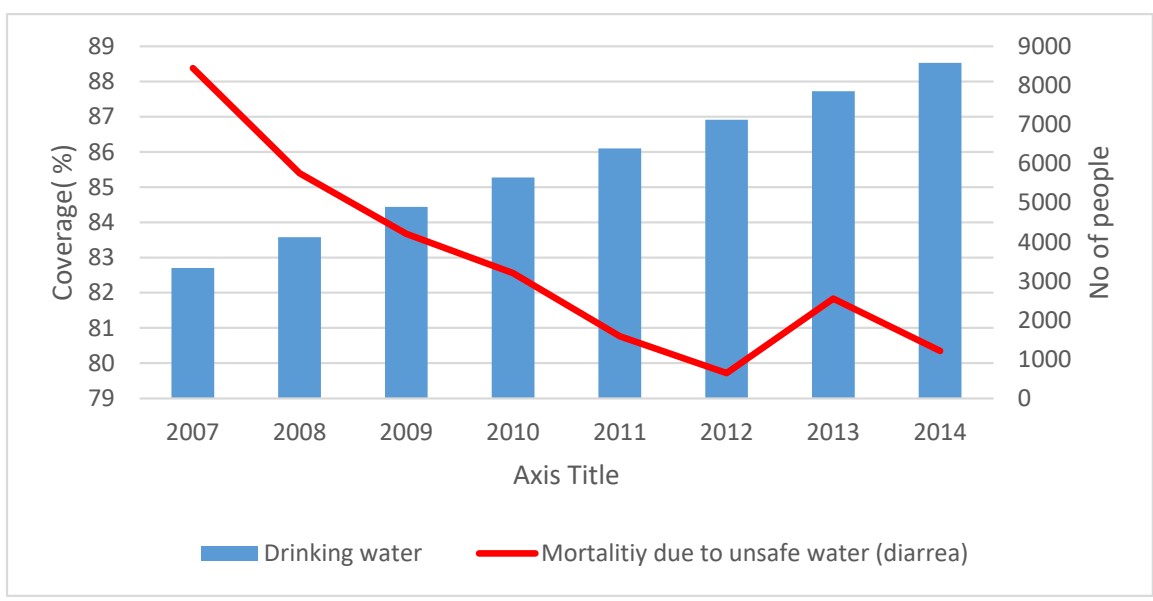

**Figure 3.** Graph of indicators 3.9.2 and 6.1.1 in Indonesia (Sources: statistic data from JMP and BPS).

The successful synergy between water and health causes good integration of the related targets, as shown in Supplementary Table S1. According to the standard coefficient of indicator 6.1.1, the number of deaths due to unsafe water and sanitation (3.9.2)) was reduced to −0.68; the coefficient of this indicator was higher than those of the GDP (0.04), and population (0.08) (in Supplementary Table S1). If countries commit to increasing their numbers of drinking water services, population sizes and GDPs will increase, and the number of deaths due to unsafe water and sanitation will decrease.

### 3.1.3. Energy- and Water-Related SDG Nexus

Rapid population growth and economic development enhance water and energy demands and create excessive challenges to achieving SDGs. These challenges are partly because energy and water usage are inseparably associated with manufacturing and operating activities. Supplementary Table S2 shows that access to electricity and sustainable modern energy services underpins health, education, and livelihoods. Hence, the targets of energy-related SDGs show high synergies and trade-offs with other targets, including targets of water-related SDGs. The total number of synergies and trade-offs interlinkages between energy-related targets and other targets were 9 and 7, respectively. Indicators 7.1.1 (access to electricity) and 7.1.2 (reliance on clean fuels and technology) are synergistically related to indicators of health (3.3.5 and 3.9.2), education (4.a.1a/b), and water (6.1.1). However, indicator 7.2.1 (renewable energy) shows a trade-off with some water-related indicators, i.e., 3.9.2, 6.1.1, 7.1.1, and 7.1.2 (in Supplementary Table S2).

Renewable energy is an essential factors in developing rural areas and guaranteeing energy availability in the future. However, fossil fuels still dominate energy use, accounting for approximately 80% of the global demand [67]. Moreover, the price of fossil fuels is lower than that of renewable energy; for example, in Indonesia, the government provides a subsidy for the community and derives a raw fossil fuel stock from the earth. However, the mining of raw fossil fuels has been decreasing. In 2018, the target production was 800 barrels/day; however, the current amount produced was lowered to 773 barrels/day. Production appears to be decreasing, as the average production in 2017 was 935 barrels/day (BPS). Hence, the Indonesian government has shifted to renewable energy in response to increases in energy demand and population growth. The limited technology, proper conversion facilities and high cost of installing and maintaining renewable energy facilities cause a trade-off with other indicators. Moreover, there is a lack of support from national policies for renewable energies; for instance, regulation No. 49/2018 of the Ministry of

Energy and Mineral Resources states that as the leading company in producing electricity, State Electricity Enterprise (PLN) will pay only 65% of the solar energy production costs that other companies pay, making it impossible to cover the production cost.

The synergies between water- and energy-related indicators was caused by the interlinkage of goals, as shown in Supplementary Table S1. The standardized coefficient of energy was higher than those of GDP and population. For example, the results of the correlation between 6.1.1 (a dependent variable) and 7.1.1 (an independent variable) show that the adjusted $R^2$ was 0.7 and the standardized coefficient of the indicator 7.1.1 was 0.71. The energy conversion chain, which includes the resource utilization process, power plant cooling, and system operation maintenance, requires water. Conversely, energy is required for the withdrawal, collection, processing, and distribution of water. These interlinkages support the finding of trade-offs between these sectors.

### 3.1.4. Research Development and Waste Treatment

Advanced technologies and research may support quality of life. Waste management and technologies improve living standards. However, significantly growing populations produce increasing amounts of waste annually. Domestic material consumption increased by up to 92 billion metric tons globally, and 6 billion metric tons for sub-Saharan Africa and Northern Africa in 2017, despite limited natural resources [68]. This material consumption leads to an increase in waste for all nations.

Moreover, increasing community size creates considerable risks and challenges to achieving SDGs. Human activities results in various pollutants entering water sources [24]. Water pollution mostly stems from domestic, agricultural, and industrial waste. Increasing the population size will increase the consumption of goods that do not decompose easily, such as electronic waste (e-waste) and hazardous waste. Currently, almost everybody uses a phone and battery in their daily life. However, the facilities for recycling these items are limited in all nations, especially in low- and middle-income countries. E-waste, which contains over 60 hazardous materials (e.g., lead, mercury, cadmium, and beryllium) poses a severe threat to the pollution of the environment and population [69]. The World Bank projected that by 2025, domestic waste would reach 1.8 million tons [70] and the annual growth of electronic waste would reach 50 million tons [71]. Hence, coordinating advanced waste treatment technologies (12.4.2) and continuous research development (9.5.1) will help nations achieve water-related SDGs.

Research development (9.5.1) and waste treatment (12.4.2) showed good synergy ($\rho = 0.52$), as shown in Supplementary Table S2, indicating that advancements due to research findings escalated the amount of waste treated. As shown in Supplementary Table S1, the interlinkage between these indicators was caused by the correlation between them. When indicator 12.4.2 was used as the dependent variable, indicator 9.5.1 was the most important variable, with a standardized coefficient of 0.46, followed by the indicators of population and GDP. Researchers have been working to find a solution to waste treatment. For example, Monteiro 2016 utilized seafood waste to remove hazardous substances ($Hg^{2+}$ and $Cd^{2+}$ ($\mu g/L$)) from water [72]. Payus 2019 developed an efficient used of durian husks to remove physical pollutants from groundwater supplies [73].

### 3.2. Interlinkages between Water-Related SDG Indicators in Low-Income Countries (LIC)

Sustainability is recognized as the most exciting and challenging concern of this era [74], and it is much more challenging in low-income countries than in middle- and high-income countries. As Folke et al. [75] described, SDG outputs are structured as three layers of a wedding cake; those layers are economy, society, and biosphere, where the economy (the top layer) serves the needs of society, and the biosphere acts as the foundation. This structure makes it challenging for low-income countries to achieve the SDGs.

The available data from low-income countries are limited, as shown in Figure 4. Health, water, and energy were the most frequent themes in the low-income countries. Health was influenced by the ability to access clean water and energy.

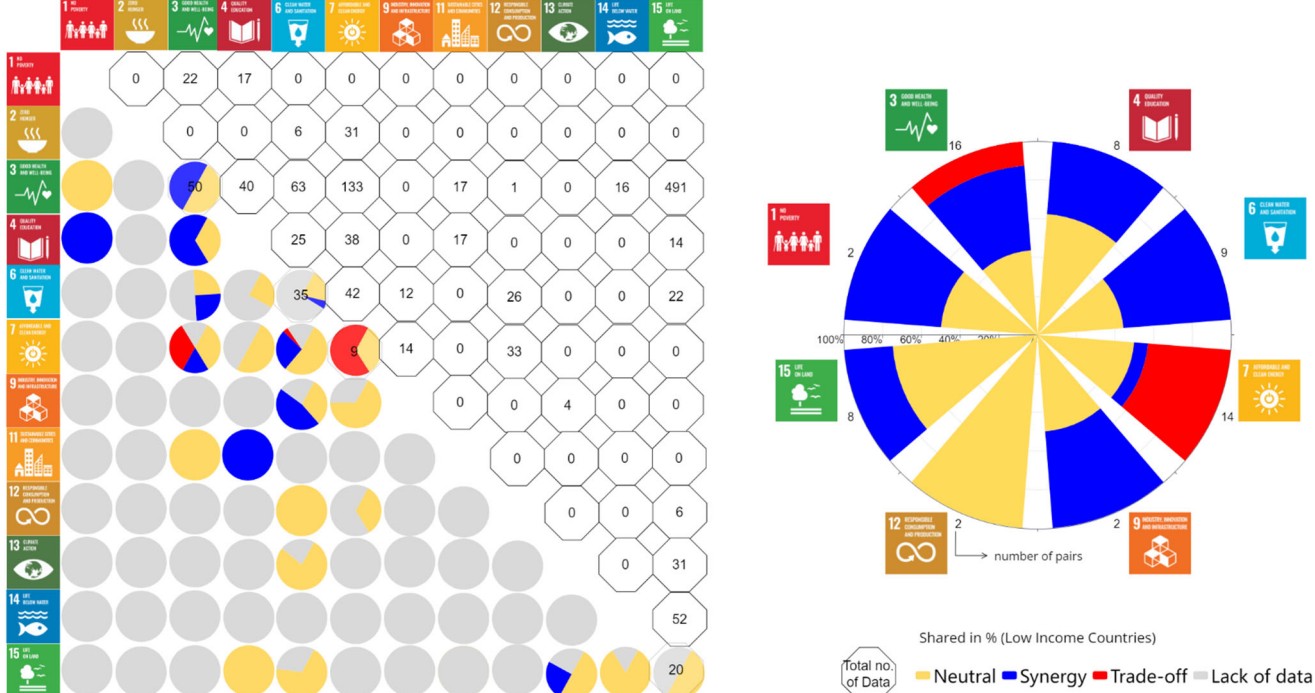

**Figure 4.** Matrix of interlinkage percentage with number of samples (**left**) and shares in percentage of interlinkage with total number pairs in LIC (**right**).

Global countries show more interlinkages (synergy) than LIC. It is caused by the influence of developed countries. Many developed countries achieved more goals in SDGs than developing countries; for instance, the Netherlands established policies to achieve some SDG targets by 2020. Successful implementation of SDGs in the Netherland is the result of well-organized national and local policies and stakeholder collaboration [76]. However, limitations faced by low-income countries make it difficult to achieve SDGs, and progress toward SDGs is difficult to measure due to the challenges of LIC submitting their progress to the UN SDG database. As shown in Figure 4, only 61 pairs of interlinkages from in LIC are available for analysis. The synergies and trade-offs between indicators globally and for low-income countries are mostly similar. However, some neutral interlinkages appear in low-income countries, especially among NTDs (3.3.5), drinking and hand washing facilities at schools (4.a.1), changes in water use (6.4.1), and the proportions of the population with access to electricity (7.1.1). These indicators show synergistic or trade-off interlinkages in the global analysis, but fewer connections (i.e., neutral associations) in LIC. This might happen because policies in the analyzed countries might focus on only one target. Hence, we did not find a significant trend in the synergistic or trade-off interlinkages.

### 3.3. Mortality Due to Unsafe Water and Sanitation in Low-Income Areas

Concentrating on the water and sanitation requests of people living in informal urban settlements will be necessary to assure comprehensive success of the SDGs. However, compared to the global results, the results of the low-income countries indicate that these countries face severe water resource and quality challenges due to their inadequate water resources among other issues. According to the results of the data analysis, the mortality due to unsafe water globally and in the low-income countries decreased from 1990 to 2017 by up to 4.1% and 3.7%, respectively (S7). Water access crises are a substantial health problem, with dysentery and the starvation of children under five years of age representing particularly dire issues [64,77]. The results of this study proved that increased water access and treated drinking water affect the health of populations, especially in low-income countries, as shown by the interlinkages (synergies and trade-offs) of indicator 3.9.2 (mortality due to unsafe water and sanitation), which has the highest number of interlinkages among

the low-income countries (Supplementary Table S2). Supplementary Table S2 shows that indicator 3.9.2 had synergies with indicators 3.3.3 (malaria), 4.a.1b (schools with basic hand-washing facilities), 6.1.1 (drinking water), 7.1.1 (access to electricity), and 7.1.2 (reliance on clean fuels and technologies) ($\rho = 0.41$, $\rho = -0.46$, $\rho = -0.76$, $\rho = -0.66$, and $\rho = -0.56$, respectively), and a trade-off with indicator 7.2.1 ($\rho = 0.51$).

Attempts to improve access to safe drinking and sanitation are necessary to achieve water-related SDGs. The mortality rate due to unsafe water and a lack of sanitation and hygiene (3.9.2) decreased, as shown in Supplementary Table S2. Prüss-Ustün 2019 mentioned that mortality in low- and middle-income countries due to inadequate water, sanitation, and hygiene decreased by 17,000, 152,000, and 132,000 deaths, respectively, from 2012 to 2016 [78]. Although deaths due to unsafe water and a lack of sanitation and hygiene decreased, the number of daily diarrheal cases was still high in 2016 (up to 49.8 million cases), accounting for 829,000 deaths in low- and middle-income countries [78]; for sub-Saharan Africa (SSA), unsafe water and the lack of sanitation and hygiene accounted for 7.75% of the total deaths due to diarrhea [79]. Access to sanitation facilities is a problem in the city of Harare, Zimbabwe. Even though sanitation access has been improved by up to 80.6%, more than 253,000 people still lack access to sanitation facilities. Moreover, shortage of water and sanitation destabilizes productivity and economic development [80].

Dams, which serve as drinking water and energy sources, provide energy, and clean water. Africa has seen a resurgence in dam construction, including the installation of 980 large dams to provide more improved water sources and create hydroelectricity, particularly in Ethiopia, which has overflowing rivers and mountainous areas [81]. Dams can provide energy sources for power plants and health risks (e.g., malaria). Moreover, a side effect of dams is high population density, which ranges from as low as 1.2 people/km$^2$ to as high as 2478 people/km$^2$ near dams; the median population density calculated for each relevant WHO subregion ranges from 25.8 people/km$^2$ in WHO subregion 2 to 764 people/km$^2$ in WHO subregion 7 [82]. High population density with insufficient sanitation and hygiene creates health problems. Additionally, the water source and water discharge outlet are located in the same place. Hence, a trade-off may occur due to the lifestyle of the community surrounding a dam. Furthermore, the obstacles to clean water and sanitation in low-income countries include economic and spatial issues, social exclusion, institutional and political issues and decision-making, and insufficient data [83]. Hence, a trade-off may occur between mortality due to unsafe water and a lack of sanitation and hygiene (3.9.2) and renewable energy (7.2.1).

## 4. Discussion

This study investigated the qualitative judgment of interlinkage among SDGs from the current data of the UN. The limitation of funding and resources for all nations becomes an obstacle to achieving all the SDGs, especially water-related SDGs. On the other hand, all sectors demand water. Hence, this study provided feasible suggestions to global countries and LIC in prioritizing the achievement of specific SDGs, especially water-related SDGs. As mentioned in the introduction, several studies were conducted to calculate the interlinkage among SDGs. All the studies mentioned the interlinkage among SDGs that showed synergy outweighs synergies [5,10,84,85]. The same finding in this research found that interlinkage between water-related SDGs shows more synergies than trade-off for global and LIC analysis. Moreover, the previous studies mentioned the variation in interlinkage due to factors (i.e., population, regional, and income) [10]; however, the strong driver causing the interlinkage was not clearly mentioned in the previous study. Thus, this study investigated the strong driver in causing interlinkage (synergies and trade-off), and, moreover, this study suggested the priority key in achieving SDGs related to water.

### 4.1. Priority Keys in Water-Related SDGs

Figure 2 shows that goals related to water (goal 6), energy (goal 7), health (goal 3), education (goal 4), and consumption and production (goal 12) are the primary goals of

water-related indicators for achieving SDGs globally. These goals indicate that achieving these indicators boosts the success of SDGs related to water globally. This study categorized the water-related SDGs into seven keys (i.e., a. extreme water events, b. water availability, c. water quality and waterborne disease, d. water-related to energy, e. water-related to industry and technology, f. water governance and management, and g. water-related to ecosystems). This study may help the government prioritize the target and find the most affected categories to achieve SDGs.

Figure 5 shows the expert judgment based on the qualitative measurement of SDGs in this study. We categorized the indicators into seven keys, calculated the number of interlinkage (neutral, synergy, and trade-off), and threshold by using equal interval with three classes (high, medium, and low). Hence, in this study, we strongly advise global (all nations) and LIC to focus on water quality and waterborne disease. Many studies have demonstrated that inadequate clean water and sanitation facilities increase health problems, especially those related to skin and eyes, gastrointestinal illnesses, and malaria, responsible for deaths annually [86]. Furthermore, water supply and home sanitation are correlated with physical and mental health [87]. Moreover, the novel coronavirus, SAR-CoV-2, which causes COVID-19, has increased the need for people globally to wash their hands with soap. Thus, people need more clean water to prevent virus transmission [88]. Additionally, water availability in the global community, and water availability globally (all nations) is higher than in LIC. It proves that, even though water availability in LIC has improved, the LIC average is still lower than the average global. People in LIC spend time and energy accessing water since less water sources in or near home [89]. It continues to challenge the LIC government to work more on providing water to their people.

Moreover, the government in global (all nations) should be more careful in developing alternative energy to replace fossil energy, especially when utilizing water as the primary energy source. Water-related energy shows trade-offs in global analysis and more trade-offs in LIC. The problem comes in developing countries when the government only focuses on energy resources without considering the side impact of energy resources. As mentioned in previous studies, developing a dam may protect people from drought, but in the dry season, the dam's water level should be well maintained; otherwise, the dam turns into a breeding pool for malaria [82]. Moreover, a dam may increase agriculture production; however, on the other hand, conflict may arise among the community due to water utilized only for a specific sector and sacrifice the other sectors, such as household and industry [24,90,91]. A dam may become a center of activity that allows the community to access and discharge water from and to the same water inlet/outlet, which may cause health problems, especially in LIC, where the sanitation facility is insufficient. Moreover, a dam is designed to be flood protection; however, some dams fail in raining season and cause devastating floods [92,93].

Limitations of human resources and slow economic growth are obstacles for LIC in developing technology. It caused the technology gaps between High Income Country (HIC), Middle Income Country (MIC), and LIC. Another challenge for LIC is water governance and management, even though the physics of water resources are available in LIC. The social condition and political factors of water governance induced discrimination of water access and sanitation services in LIC, whereas the economic factor of water governance strongly inclined the disproportion of water access [94].

Water-related disasters and water-related ecosystems are weak in achieving water-related SDGs both in global and LIC. It proves that even global countries (including HIC and MIC) are still struggling to reduce water-related disaster impact. Hirabayashi et al. predicted that severe flood events would occur in the future, and flood impacts humans and finances severely, even in high-income counties [95,96]. as Additionally, the water-related ecosystem will suffer, due to the demand of life, as people sacrifice the nature to fulfill their demands, and forests are vulnerable to human activity; for example, converting natural forests into industrial forest, settlements [90,91], etc., that risk water potential because forests influence stream discharge, precipitation, evapotranspiration

(ET), infiltration, groundwater recharge, runoff, and water discharge to streams, which are central components of the hydrological cycles [97].

**Figure 5.** The expert judgment from percentage of interlinkage among indicators of water-related SDGs in global and low-income countries (LIC).

As shown in Table S1, the interaction between indicators is more prominent than GDP and population drivers to achieve synergies and avoid trade-offs. It is crucial to consider the relationship between indicators when government focuses on achieving SDGs. For example, a school with a hand washing facility (4.a.1b) shows synergy with a population with drinking water (6.1.1). If the communities have good access to drinking water, they may be able to provide water for sanitation at school.

*4.2. Future Study*

U.N. Agenda 2030 is an ambitious target. Some indicators analyzed in this study, e.g., agricultural areas under productive and sustainable agriculture (2.4.1), and cities with a direct participation structure of civil societies in urban planning (11.3.2), are not available for all nations. Hence, this study only analyzed 31 indicators from 33 selected indicators due to data limitation; only a limited number of countries report their data to the UN because the SDG targets are not included in the policies of all countries. This creates an obstacle to measuring the achievement of and interlinkages between the SDGs at the national level. Moreover, the SDGs is one of method to achieve sustainability in the future; thus, to measure the achievement of SDGs in the future (for example, the end of 21st century), future analysis is required. Numerical modeling is a potential solution to filling the gap in the U.N. data. Some prominent data sources can be utilized to address the gaps in this study, i.e., the numbers of deaths and missing people (1.5.1, 11.5.1, 13.1.2) [95], and water stress (6.4.2) [98]. The benefit of numerical modeling is that the data cover a large range of years and are available for certain scenarios. Moreover, models can be applied to support mitigation and adaptation at the beginning and end of the 21st century.

**5. Conclusions**

Spearman's rank correlation measured water-related SDG indicators globally and in LIC. The key priorities for achieving water-related SDGs are water-related health and water quality to support life at home and school (goals 3, 4, and 6). Moreover, the government should pay attention when focusing on achieving goal 7, especially for countries that utilize water to provide an alternative energy resource. Goal 7 showed high synergies and trade-offs globally, particularly in LIC.

Globally, achieving water- and energy-related goals improved health and education, as shown in Table S2. Reducing mortality due to malaria, NTDs, and unsafe water and a lack of sanitation can be achieved by providing safe drinking water. However, water sources should be protected from harmful activities that pollute water sources and create

new problems. The results of the Spearman's correlation analysis showed a trade-off between renewable energy and water-related diseases (e.g., malaria, NTDs, and diseases due to unsafe water and a lack of sanitation (Supplementary Table S2). For example, it has been believed that dam construction benefits communities, providing water for multiple purposes, supporting power generation and recreation, and protecting against damaging floods. However, dam maintenance is challenging, and dams can increase mortality due to a lack of safe water, sanitation, and hygiene (indicator 3.9.2). The construction of 980 large dams in Africa contributed to the occurrence of 1.1 million malaria cases; it has been predicted that in 2050, the number of malaria cases would reach 2 million due to increasing population size and climate change. Furthermore, settlements around dams with improper sanitation and hygiene facilities pose health risks. In remote areas of low-income countries, water is used and discharged from the same location. This behavior generates health problems. These issues need to be mitigated because of the risks of reservoir-driven, water-related diseases. Improved maintenance, education, and policies for battling diseases in the vicinity of reservoirs are required to achieve the SDGs, especially water-related SDGs.

Furthermore, the findings of this study suggest that the increasing population size intensifies domestic consumption and waste globally and in low-income countries. However, there is a synergistic interaction between waste treatment and research development, as shown in Supplementary Table S2.

Water-related SDG indicators are challenging to achieve. Of the 231 SDG indicators, 33 are associated with water-related SDGs. The relationships between indicators are the primary drivers of the interlinkages between indicators. As shown in Supplementary Table S1, strong interlinkages were found between 34 pairs of indicators due to their relationships; the interlinkages between 11 indicators were driven by GDP, and the interlinkage between one pair was driven by population. Hence, the global (all nations) and LIC governments have to be more careful in choosing SDGs target to achieve since it may affect synergy or trade-off to other goals.

**Supplementary Materials:** The following supporting information can be downloaded at: https://www.mdpi.com/article/10.3390/w15040613/s1: Supplementary Figure S1.(a) Positive trend; (b) negative trend; Supplementary Figure S2 shows the trends of example indicators (3.3.5 and 6.4.1) for different income levels and a scatterplot of data related to indicators 3.3.5 and 6.4.1 for all nations; Supplementary Figure S3 Scatter plot of the correlations between indicators, GDP and population size; Supplementary Figure S4 Scatterplot shows yearly trend (the increase/decrease) of indicator influenced by population and GDP; Supplementary Table S1 The magnitudes of the standardized coefficients (beta) of synergistic and trade-off interactions between indicators; Supplementary Table S2 Spearman's correlation coefficients and the number of samples for global and low-income country interlinkages.

**Author Contributions:** A.B.R.: conceptualization, methodology, investigation, writing—original draft; Y.H.: supervision, methodology, writing and reviewing. All authors have read and agreed to the published version of the manuscript.

**Funding:** The research was supported by the Environment Research and Technology Development Fund (JPMEERF20202005) of the Environmental Restoration and Conservation Agency of Japan.

**Institutional Review Board Statement:** Ethical approval for this type of study is not required by our institute.

**Informed Consent Statement:** Not applicable.

**Data Availability Statement:** All the data from this study can be downloaded for free from the official homepages of the U.N. Statistics Division and World Bank.

**Acknowledgments:** We gratefully acknowledge our financial support from the Ministry of Environmental Japan, and nonfinancial support from the Shibaura Institute of Technology. We also wish to thank the anonymous reviewers of this manuscript.

**Conflicts of Interest:** The authors declare that they have no known competing financial interest or personal relationship that could appear to influence the work reported in this study.

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
