# Peer review of "Interlinkages of Water-Related SDG Indicators Globally and in Low-Income Countries"

_water, doi:10.3390/w15040613_

Round 1

Reviewer 1 Report

This study examined the “Interlinkages of water-related SDG indicators globally and in low-income countries”. Given that water sustainability is a global concern, this paper is timely and could offer new insights on Interlinkages between water-related SDG indicators. The manuscript is generally well written and easy to understand. I suggest that the authors revise the manuscript incorporating the following comments and suggestions into an updated version.

1.       Did the authors normalize the data before statistical analysis?

2.       It is difficult to consider that a linear statistical approach for all complex factors (water-related SDG indicators) could be sufficient.

3.       Why the authors considered only low income countries excluding the high income countries. It would be much better if they inter-compare the above mentioned categories of countries.

4.       The study area map must have map elements i.e. scale, north arrow and legends

5.       Improve the quality of figures

Author Response

Thank you very much for your valuable comments and feedback, we have revised the paper according to your input and feedback.

  1. Did the authors normalize the data before statistical analysis?

Thank you very much for this important question, which we explained in lines 222-227

  1. It is difficult to consider that a linear statistical approach for all complex factors (water-related SDG indicators) could be sufficient.

We agree, hence, we compared the 3 prominent variables in Table S5 (supplementary).

  1. Why did the authors consider only low-income countries excluding the high-income countries? It would be much better if they inter-compare the above-mentioned categories of countries.

Thank you very much for your suggestion; we explained in lines 117-120

  1. The study area map must have map elements i.e., scale, north arrow, and legends

Thank you very much. We revised the map, since the map just shows the country locations (using a small scale), we don’t input a lot of information (such as coordinates, mapping data, or projection system) because we will not use any physical measurement, a lot of map element will cover the main information of map. Moreover, we did not add the scale bar information to coordinate system WGS84, distortion shape in the Antarctic. However, we added a north arrow. Thank you for your feedback about the cartographic elements.

  1. Improve the quality of figures

Thank you very much we have revised the figures

Reviewer 2 Report

Title: Interlinkages of water-related SDG indicators globally and in low-income countries

Dear authors,

The topic of your paper is interesting. Overall, the quality of this paper is adequate, however there are some aspects which should be improved.

Following are some recommendations to consider:

A.      Abstract, title and references:

The aim is clear.

The references are relevant, recent, and appropriate.

Please check the use of references and citations in the text. I am not sure it's the right one, on the other hand, it's not homogeneous: i.e. line 359-360: "poses a severe threat to the pollution of the environment and population (de Oliveira Neto et al., 2017)."

In recent years, there are some researches water uses and impact on SDG (specially about Cat. B water availability and Cat. F water governance and management). I think some of theme should be included. Some of these studying have been published even in this same journal, for example:

-          Pérez, D.M.G.; Martín, J.M.M.; Martínez, J.M.G.; Sáez-Fernández, F.J. An Analysis of the Cost of Water Supply Linked to the Tourism Industry. An Application to the Case of the Island of Ibiza in Spain. Water 202012, 2006. https://doi.org/10.3390/w12072006

The next one is also interesting, very related with your reference number 54:

-          González-Pérez, D.M; Martín, J.M.; Guaita, J.M.; Morales, A. Analyzing the real size of the tourism industry on the basis of an assessment of water consumption patterns. Journal of Business Research 2023, 157, 113601. https://doi.org/10.1016/j.jbusres.2022.113601

B. Introduction:

The research question is clearly outlined. However, I would suggest to argue more why this study may contribute with new knowledge. Please, clarify why your paper is important. Who is going to earn more with your results about Interlinkages of water-related SDG indicators.

C. M&M:

The variables are defined and measured appropriately. Methods are valid and reliable.

D.- Results & Discussion:

Data in appropriate way. Figures and tables are ok. Discussion is supported by results or/and references.

However, some figures and their size must be edited so that the details collected in them are visible. For example, figure 2: is not 100% visible. The details that are not visible

E.- Conclusions/ Implications:

Conclusions are supported by results or/and references. However, there is some aspects which should be improved: suggestions for future research.

It would be interesting some additional managerial implications in line with the findings of the study. Practical implications?  Something to inspire future research or implications for practice.

Kind regards,

Author Response

Thank you very much for your valuable comments and feedback; we have revised the paper according to your input and feedback.

  1. Abstract, title, and references:

The aim is clear.

The references are relevant, recent, and appropriate.

Please check the use of references and citations in the text. I am not sure it's the right one, on the other hand, it's not homogeneous: i.e. line 359-360: "poses a severe threat to the pollution of the environment and population (de Oliveira Neto et al., 2017)."

In recent years, there are some researches water uses and impact on SDG (specially about Cat. B water availability and Cat. F water governance and management). I think some of theme should be included. Some of these studying have been published even in this same journal, for example:

-          Pérez, D.M.G.; Martín, J.M.M.; Martínez, J.M.G.; Sáez-Fernández, F.J. An Analysis of the Cost of Water Supply Linked to the Tourism Industry. An Application to the Case of the Island of Ibiza in Spain. Water 202012, 2006. https://doi.org/10.3390/w12072006

The next one is also interesting, very related with your reference number 54:

-          González-Pérez, D.M; Martín, J.M.; Guaita, J.M.; Morales, A. Analyzing the real size of the tourism industry on the basis of an assessment of water consumption patterns. Journal of Business Research 2023157, 113601https://doi.org/10.1016/j.jbusres.2022.113601

Thank you very much for the recommended reference; we added these valuable references to enrich the value of this manuscript.

  1. Introduction:

The research question is clearly outlined. However, I would suggest to argue more why this study may contribute with new knowledge. Please, clarify why your paper is important. Who is going to earn more with your results about Interlinkages of water-related SDG indicators.

Thank you very much; we emphasized the government may take advantage of this study; it is written in the abstract and introduction (lines 51-21; 98-100)

  1. M&M:

The variables are defined and measured appropriately. Methods are valid and reliable.

 Thank you very much for your judgment as an expert in evaluating the method

D.- Results & Discussion:

Data in appropriate way. Figures and tables are ok. Discussion is supported by results or/and references.

Thank you very much, we are appreciated your evaluation

However, some figures and their size must be edited so that the details collected in them are visible. For example, figure 2: is not 100% visible. The details that are not visible

We have revised the figures

E.- Conclusions/ Implications:

Conclusions are supported by results or/and references. However, there is some aspects which should be improved: suggestions for future research.

It would be interesting some additional managerial implications in line with the findings of the study. Practical implications?  Something to inspire future research or implications for practice.

Thank you very much; we added future research. We added to section 4.2

Reviewer 3 Report

The article submitted for review is not suitable for publication in its current form. It doesn't bring anything new. It is poorly written and incomprehensible. Research methodology and description of results are inconsistent and chaotic. The discussion section is not a discussion of the research results but a summary. conclusions are not specific. They are too general and can be written without results and analysis of calculation results. It requires a complete reorganization and rethinking. It's too long and too general. Reduce the number of references and focus on the most recent reports from the last 5 years. Not suitable for publication in its current form. This is not a well thought out and well written article.

Author Response

Thank you very much for your valuable evaluation to enrich this manuscript. We are sorry for the previous lengthy manuscript. We have revised the manuscript according to the reviewers’ comments. Moreover, we shorten the manuscript to increase its readability, and we removed old references and only refer to the up-to-date references.

Round 2

Reviewer 3 Report

The authors have significantly changed the structure of the article. They made the recommended changes and greatly improved the understanding of the content and purpose of the publication. I have no comments on the current form of the article.